# Treatment Outcomes and Significance of Multimodal Treatment in Esophageal Squamous Cell Carcinoma with Synchronous Oligometastasis

**DOI:** 10.3390/cancers17213407

**Published:** 2025-10-23

**Authors:** Manato Ohsawa, Yoichi Hamai, Yuta Ibuki, Tomoaki Kurokawa, Nao Kitasaki, Morihito Okada

**Affiliations:** Department of Surgical Oncology, Hiroshima University, Hiroshima 734-0037, Japan; ohsawa311@gmail.com (M.O.); ahryyibuki@gmail.com (Y.I.); tomokanakurokawa@yahoo.co.jp (T.K.); tennis.xylitol@gmail.com (N.K.); morihito1217@gmail.com (M.O.)

**Keywords:** esophageal squamous cell carcinoma, prognosis, oligometastasis, surgery, chemoradiotherapy, chemotherapy, immunotherapy, radiotherapy

## Abstract

Esophageal squamous cell carcinoma is an aggressive cancer that often spreads to distant sites, making treatment difficult and survival rates poor. However, some patients have only a limited number of metastases, a condition known as oligometastasis. This situation may represent an intermediate state in which local treatments, such as surgery or radiotherapy, could still provide benefits, especially when combined with systemic therapies. In this study, we reviewed the outcomes of patients with esophageal squamous cell carcinoma and synchronous oligometastasis who were treated with different approaches, including systemic therapy alone, local therapy alone, or a combination of both. Some patients who received combined systemic and local therapies achieved favorable outcomes, whereas multiple organ metastases were associated with extremely poor prognosis. These findings suggest that patients with oligometastatic disease may benefit from aggressive multimodal treatment, challenging the view that stage IVB disease is always incurable.

## 1. Introduction

Esophageal cancer is the seventh-most common cancer worldwide and the sixth leading cause of cancer-related deaths, with an overall poor prognosis. Squamous cell carcinoma is the predominant histological type, accounting for approximately 90% of all cases [1,2]. For resectable advanced esophageal cancer, multimodal treatment involving surgery, chemotherapy, and chemoradiotherapy is applied depending on the stage of progression [3,4]. In contrast, unresectable esophageal cancer is classified as stage IV according to the tumor–node–metastasis classification, eighth edition, with stage IVB being particularly difficult to treat [5]. The main treatment for stage IVB is systemic drug therapy; however, the 5-year survival rate is only 5%, highlighting the urgent need for therapeutic improvements [6,7].

In cases of localized recurrence, local therapy, such as surgery or definitive chemoradiotherapy, may be pursued instead of systemic drug therapy. First introduced in 1995, oligometastasis describes a condition characterized by a few distant metastases. It has become an area of clinical interest, representing an intermediate state between localized and systemic metastases. Oligometastases reflect unique tumor biology, and local treatment can achieve long-term disease control or even cure in some patients [8,9]. Several studies on oligometastatic recurrence of esophageal cancer have shown that patients respond to local treatment, potentially improving overall survival [10,11,12,13].

Cases diagnosed with oligometastasis at initial presentation are referred to as synchronous oligometastases and classified as stage IVB. However, this presentation may also represent an intermediate state characteristic of oligometastasis in which local treatment may be effective. Improving the prognosis of patients with synchronous oligometastases, such as esophageal squamous cell carcinoma (ESCC), requires evaluation of treatment outcomes. However, few comprehensive reports have examined treatment outcomes in this patient population. In addition, the concept of oligometastasis in ESCC remains controversial, and no universally accepted definition exists; therefore, further data are required. This study aimed to summarize treatment outcomes in patients with ESCC with synchronous oligometastasis and to examine the significance of multimodal treatment, including local therapy.

## 2. Materials and Methods

### 2.1. Patients

A total of 191 consecutive patients with ESCC and synchronous oligometastasis, diagnosed at the initial examination and treated at Hiroshima University Hospital between October 2006 and December 2022, were included in this study. Oligometastasis was defined as five or fewer distant metastatic lesions. Table 1 summarizes the patients’ clinicopathological features. Tumors were staged according to the *TMN Classification of Malignant Tumors, 8th edition* [5]. Cases diagnosed before 2016 were reclassified according to the 8th edition. Clinical responses to chemotherapy, radiotherapy, and chemoradiotherapy were evaluated using the Response Evaluation Criteria in Solid Tumors [14]. Accordingly, the primary tumor was an unmeasurable lesion, and treatment response was evaluated based on measurable lesions among lymph node metastases and distant metastases. Performance status was assessed using the Eastern Cooperative Oncology Group performance status (ECOG PS) scale. Patients’ clinical data were obtained from databases and medical records. The Institutional Review Board of Hiroshima University approved the study protocol (approval number: 2225) and waived the requirement for informed consent because of the retrospective design.

### 2.2. Oligometastasis

Oligometastasis was defined as five or fewer distant metastatic lesions. For distant lymph node metastases, each station was counted as one metastatic lesion regardless of the number of lymph nodes (e.g., two lymph node metastases around the abdominal aorta were counted as one lesion). The number of metastatic organs was not included in the criteria for oligometastasis; however, when assessed, distant lymph node metastasis was regarded as single-organ metastasis, even if multiple stations were involved.

Diagnosis of oligometastasis was comprehensively assessed using computed tomography (CT) and ^18^F-fluorodeoxyglucose–positron emission tomography. Magnetic resonance imaging was also performed in cases of bone or liver metastases. Two independent radiologists evaluated all diagnostic images. When imaging findings alone were insufficient for diagnosis and biopsy was feasible, such as in bone metastasis, a biopsy was performed.

The sites of distant metastasis were the distant lymph nodes, liver, bone, lungs, adrenal glands, and skeletal muscle. The stations for distant lymph node metastasis are shown below. The cervical region included lymph nodes of the deep cervical, cervical pretracheal, and supraclavicular areas. The thoracic region included lymph nodes in the thoracic pretracheal, posterior thoracic paraaortic, anterior mediastinal, ligamentum arteriosum, hilar, right tracheobronchial, and axillary areas. The abdominal region included lymph nodes around the abdominal aorta, on the posterior surface of the pancreatic head, along the common hepatic artery (posterior group), within the hepatoduodenal ligament, and in the mesentery of the small intestine.

### 2.3. Treatment Selection

Surgeons, radiologists, and oncologists jointly discussed the treatment plan for each patient. Based on the patient’s condition, the extent of distant metastasis, and the resectability of distant lesions after initial therapy, antitumor treatment, such as chemotherapy (including immunotherapy), radiotherapy, surgery, or appropriate combinations of these modalities, was recommended.

Although systemic drug therapy, including chemotherapy and immunotherapy, was the primary treatment of choice, patients with acceptable general condition and comorbidity profiles could also receive more intensive treatments, such as chemoradiotherapy or curative surgery. When surgery was feasible, it was preferred over chemoradiotherapy. For organ metastases, cases where organ metastases disappeared on imaging after preoperative therapy were considered suitable for surgery. For distant lymph node metastases, cases involving metastases in areas relatively close to the esophagus (such as lymph nodes in the thoracic pretracheal region) were considered suitable for surgery.

Surgery was generally performed after preoperative therapy, which included either chemotherapy or chemoradiotherapy. Cases included conversion surgery after initial therapy (i.e., when the tumor was deemed unresectable at the initial consultation but resectable after initial therapy [15]) and planned surgery after neoadjuvant therapy. Preoperative chemotherapy was used; however, preoperative chemoradiotherapy was preferred, especially for bulky tumors. Chemotherapy and radiotherapy were difficult to administer in patients with liver or kidney dysfunction and interstitial pneumonia. Surgery alone was performed in a few cases. Patients in poor general condition or for whom chemotherapy, chemoradiotherapy, or surgery was not feasible were treated with radiotherapy alone.

A treatment plan was established according to this treatment policy. Treatment was broadly divided into three categories: systemic therapy plus local therapy (preoperative therapy and surgery, chemoradiotherapy), local therapy (radiotherapy or surgery alone), and systemic therapy (chemotherapy or immunotherapy) (Table 1).

### 2.4. Chemotherapy and Immunotherapy

Cisplatin plus 5-fluorouracil (CF), nedaplatin/5-fluorouracil, docetaxel/cisplatin/5-fluorouracil, CF/pembrolizumab, CF/nivolumab, and nivolumab/ipilimumab regimens were administered to 26 (53.1%), 7 (14.3%), 6 (12.2%), 5 (10.2%), 2 (4.1%), and 3 (6.1%) patients, respectively. The administration schedules for the CF, nedaplatin/5-fluorouracil, docetaxel/cisplatin/5-fluorouracil, CF/pembrolizumab, CF/nivolumab, and nivolumab/ipilimumab regimens were identical to those described previously [16,17,18]. For patients with impaired renal function who could not receive the CF regimen, cisplatin was replaced with nedaplatin. Chemotherapy was generally administered at standard doses, with individual adjustments made according to treatment-related toxicities.

### 2.5. Definitive Chemoradiotherapy and Radiotherapy

Chemoradiotherapy included concurrent radiotherapy (66 Gy in 33 fractions) and two cycles of CF chemotherapy, as previously described [18]. External beam radiotherapy with 10 MV X-rays was delivered concurrently in five fractions per week for 7 weeks (total dose 66 Gy). A three-dimensional treatment plan was created using a computed tomography (CT) simulator. In all cases, the irradiation field was designed to include distant metastases in addition to the basic irradiation range [19]. When concurrent chemotherapy was not feasible due to the patient’s condition, only radiotherapy was administered. The prescribed radiation doses were uniform for the whole cohort.

In cases where definitive chemoradiotherapy or radiotherapy was chosen, organ metastases included the lung, liver, and bone. Sites of distant lymph node metastases included the deep cervical, cervical pretracheal, supraclavicular, thoracic pretracheal, posterior thoracic para-aortic, ligamentum arteriosum, anterior mediastinal, hilar, axillary, para-aortic (abdominal), hepatoduodenal ligament, and small intestinal mesenteric regions. In all cases, radiation was delivered to the sites of distant metastases.

### 2.6. Preoperative Chemotherapy, Immunotherapy, and Chemoradiotherapy

CF, nedaplatin/5-fluorouracil, docetaxel/cisplatin/5-fluorouracil, and CF/pembrolizumab regimens were administered to 5 (19.2%), 4 (15.4%), 16 (61.5%), and 1 (3.9%) patients, respectively. The administration schedule was described previously [17,18]. Preoperative chemoradiotherapy and surgery consisted of concurrent radiotherapy (40 Gy in 20 fractions) and CF chemotherapy, as previously described [20,21]. External beam radiotherapy with 10-MV X-rays was delivered concurrently in five fractions per week for 4 weeks (total dose 40 Gy). Three-dimensional treatment planning was performed using a CT simulator. In all cases, the irradiation field was designed to include distant metastases to the basic irradiation field [20,21]. Preoperative chemotherapy was used; however, for bulky tumors or those with inadequate oral intake, preoperative chemoradiotherapy was preferred.

### 2.7. Surgical Treatment

Among all cases, 58 were selected for surgery. Patients underwent the following surgical procedures: open transthoracic esophagectomy (*n* = 24), thoracoscopic esophagectomy (*n* = 29), robotic esophagectomy (*n* = 5), and lymph node dissection in at least two fields (thoracic and abdominal, *n* = 16). Cervical lymphadenectomy (*n* = 42) was performed for esophageal cancer in the upper or middle third of the thoracic esophagus. Subsequently, a gastric tube was inserted to enable cervical esophagogastric anastomosis. The reconstruction route was retrosternal (*n* = 51), posterior mediastinal (*n* = 4), or anterior to the chest wall (*n* = 3). All procedures were performed by three experienced surgeons who specialize in esophageal surgery.

Surgery was selected when the only organ metastasis was to the lung. Distant lymph node metastases involved the following stations: deep cervical, supraclavicular, posterior thoracic, para-aortic, thoracic pretracheal, around the abdominal aorta, along the common hepatic artery (posterior group), and the posterior surface of the pancreatic head.

In cases where organ metastasis was present, patients underwent esophagectomy alone without resection of distant lesions when the metastatic lesions had resolved after the initial therapy and imaging tests showed no recurrence. For distant lymph node metastasis, if preoperative therapy was effective and metastases disappeared on CT, the lymph node station involved at the first examination was resected. Neck dissection was performed for supraclavicular metastasis and metastasis to the deep cervical nodes. For metastasis to the posterior thoracic para-aortic region, resection was performed using a left thoracic approach in addition to the standard esophagectomy.

### 2.8. Statistical Analysis

The results are presented as numbers (%) or medians unless stated otherwise. Comparisons between the groups were performed using an independent *t*-test. Enumeration (categorical) data were analyzed using the chi-square (χ^2^) test. Survival was evaluated using Kaplan–Meier curves and compared using the log-rank test. Progression-free survival (PFS) was defined as the time from treatment initiation to the first documented event (disease progression, recurrence, or death from any cause) or, if no event occurred, to the date of the last follow-up. Overall survival (OS) was defined as the time from treatment initiation to death from any cause or the last follow-up visit.

Multivariate Cox regression analysis was performed to identify independent predictors of OS. A backward stepwise method was used to select variables for the multivariate analysis. Statistical analysis was performed using JMP Pro 18 software (2024; SAS Institute, Cary, NC, USA), with the significance level set at *p* < 0.05.

## 3. Results

### 3.1. Details of Oligometastases and Treatments

During the registration period, there were 258 stage IVB cases, and the incidence of oligometastases was 74.0%. The incidence of oligometastases before and after the introduction of ^18^F-fluorodeoxyglucose positron emission tomography in 2008 was 61.5% and 75.4%, respectively, showing no significant difference. Table 1 shows the clinicopathological features of patients with oligometastases. Organ and lymph node metastases were present in 24 (12.6%) and 167 (87.4%) cases, respectively. The numbers of metastatic organs were one (176 cases), two (13 cases), and three (2 cases). When counting the number of organ metastases, distant lymph node metastases were regarded as one single organ. The numbers of metastatic lesions were one (120 cases), two (43 cases), three (19 cases), four (8 cases), and five (1 case). The maximum length of distant metastasis averaged 20.3 cm.

Although some cases involved multiple metastatic sites, distant metastases included distant lymph nodes (175 cases), liver (14 cases), bone (8 cases), lung (6 cases), adrenal gland (2 cases), and skeletal muscle (1 case). For distant lymph node metastasis, the involved stations included the supraclavicular (116 cases), thoracic pretracheal (37 cases), abdominal aorta (32 cases), posterior thoracic paraaortic (10 cases), anterior mediastinal (10 cases), deep cervical (8 cases), ligamentum arteriosum (4 cases), axillary (5 cases), hilar (2 cases), cervical pretracheal (2 cases), and posterior surface of the pancreatic head regions (2 cases). There was only one case for each of the other types, including metastasis along the common hepatic artery (posterior group) in the hepatoduodenal ligament, right tracheobronchial region, and small intestine mesentery.

In all patients with oligometastases, treatment was conducted according to the aforementioned policy, and the main treatment strategies were categorized into three options: systemic plus local therapy (131 cases, 68.6%), systemic therapy (49 cases, 25.7%), and local therapy (11 cases, 5.7%). No statistically significant differences were observed in baseline characteristics (ECOG PS, Clinical T, Clinical N, number of metastatic organs, number of metastatic lesions) across the three treatment strategies. Treatment-related deaths included one case following chemoradiotherapy and another following chemotherapy.

### 3.2. OS for All Cases and According to the Type of Treatment

The OS of all patients with oligometastases is shown in Figure 1A. The 3-year OS rate was 41.0%, and the median OS was 25.1 months. The OS according to the treatment period is shown in Figure 1B. The 3-year OS rates for 2006–2017 and 2018–2022 were 42.5 and 38.0%, respectively. The median OS for 2006–2017 and 2018–2022 was 27.6 and 19.5 months, respectively. No difference in prognosis was observed based on the treatment period.

The OS according to treatment type is shown in Figure 1C. The 3-year OS rates for preoperative therapy plus surgery, chemoradiotherapy, chemotherapy, surgery alone, and radiotherapy were 63.8, 32.8, 28.4, 33.3, and 14.2%, respectively. The median OS for preoperative therapy plus surgery, chemoradiotherapy, chemotherapy, surgery alone, and radiotherapy was 54.3, 20.2, 13.8, 13.2, and 5.6 months, respectively. Patients who underwent preoperative therapy plus surgery and chemoradiotherapy had significantly better OS than those who underwent other treatments.

### 3.3. OS According to Type of Distant Metastasis, Number of Metastatic Organs, Number of Metastatic Lesions, and Maximum Long Diameter of Distant Metastasis

The OS according to the factors associated with oligometastasis is shown in Figure 2. The OS rates according to the type of distant metastasis are presented in Figure 2A. The 3-year OS rates of the patients with lymph node and organ metastases were 45.0% and 13.8%, respectively. The median OS of patients with distant lymph node and organ metastases was 28.8 and 9.7 months, respectively. Patients with lymph node metastases had significantly better OS than those with organ metastasis (*p* = 0.001).

The OS rates according to the number of metastatic organs are presented in Figure 2B. The 3-year OS rates of patients with one and more than two metastatic organs were 44.3% and 0%, respectively. The median OS for patients with one and more than two metastatic organs was 28.5 and 8.6 months, respectively. Patients with one metastatic organ had a significantly better OS than those with more than two metastatic organs (*p* < 0.0001).

The OS rates according to the number of metastatic lesions are presented in Figure 2C. The 3-year OS rates for patients with one, two, and three or more metastatic lesions were 44.0, 40.0, and 26.1%, respectively. The median OS of patients with one, two, and three or more metastatic lesions was 27.5, 26.1, and 14.8 months, respectively. No significant difference was found in OS between patients with one or two metastatic lesions (*p* = 0.768); however, patients with one or two metastatic lesions had significantly better OS compared with those with three or more metastatic lesions (*p* = 0.003, 0.019).

The OS rates according to the maximum long diameters of the metastatic lesions are shown in Figure 2D. The 3-year OS rates for patients with metastatic lesions measuring 1–1.9, 2–2.9, and 3 cm or more were 44.8, 28.5, and 35.5%, respectively. The median OS for patients with metastatic lesions measuring 1–1.9, 2–2.9, and 3 cm or more was 28.5, 18.8, and 20.2 months, respectively. No significant difference in OS was found between patients with metastatic lesions measuring 1–1.9, 2–2.9, and 3 cm or more.

### 3.4. Univariate and Multivariate Analyses of OS

Table 2 presents the results of univariate and multivariate analyses of the prognostic factors for OS. Univariate analysis showed that ECOG PS (hazard ratio [HR] 1.99, 95% confidence interval [CI] 1.38–2.86, *p* = 0.0002), squamous cell carcinoma-related antigen (≥1.5 ng/mL vs. <1.5 ng/mL; HR 1.50, 95% CI 1.05–2.14, *p* = 0.024), type of distant metastasis (organ vs. lymph node; HR 2.27, 95% CI 1.39–3.68, *p* = 0.001), number of metastatic organs (≤2 vs. 1; HR 4.23, 95% CI 2.35–7.60, *p* < 0.0001), number of metastatic lesions (≤3 vs. 1; HR 2.09, 95% CI 1.29–3.38, *p* = 0.003), and type of treatment (systemic plus local therapy vs. systemic therapy; HR 0.43, 95% CI 0.29–0.65, *p* < 0.0001) were statistically significant. Multivariate analyses showed that ECOG PS (1/2 vs. 0; HR 1.52, 95% CI 1.02–2.27, *p* = 0.036), number of metastatic organs (≤2 vs. 1; HR 2.72, 95% CI 1.09–6.78, *p* <= 0.031), and type of treatment (systemic plus local therapy vs. systemic therapy; HR 0.56, 95% CI 0.34–0.93, *p* = 0.026) were statistically significant.

### 3.5. OS According to ECOG PS and Type of Treatment

The OS, according to the ECOG PS, is shown in Figure 3A. The 3-year OS rates for patients with an ECOG PS of 0 and 1/2 were 50.4% and 21.8%, respectively. The median OS for patients with an ECOG PS of 0 and 1/2 was 36.6 and 13.5 months, respectively. Patients with an ECOG PS of 0 had significantly better OS than those with an ECOG PS of 1/2 (*p* = 0.0002).

The OS according to treatment type is shown in Figure 3B. The 3-year OS rates of patients who underwent systemic plus local therapy, local therapy, and systemic therapy were 49.8, 20.0, and 20.1%, respectively. The median OS of patients who underwent systemic plus local therapy, local therapy, and systemic therapy was 35.9, 8.4, and 10.4 months, respectively. Patients who underwent systemic plus local therapy had significantly better OS than those who underwent local therapy or systemic therapy alone (*p* < 0.0001, *p =* 0.001).

### 3.6. PFS for All Cases and According to Number of Metastatic Organs and Type of Treatment

The PFS for all cases is shown in Figure 4A. The 3-year PFS rate was 21.6%, and the median PFS time was 7.1 months. The PFS rates according to the number of organ metastases are presented in Figure 4B. The 3-year PFS rates of patients with one and two or more organ metastases were 23.5% and 0%, respectively. The median PFS for patients with one and two or more organ metastases was 7.9 and 3.7 months, respectively. Patients with one organ metastasis had significantly better PFS than those with two or more organ metastases (*p* < 0.0001).

PFS according to treatment type is shown in Figure 4C. The 3-year PFS rates of patients who underwent systemic plus local therapy, local therapy, and systemic therapy were 29.3, 18.1, and 0%, respectively. The median PFS for patients who underwent systemic plus local therapy, local therapy, and systemic therapy was 9.5, 4.2, and 4.4 months, respectively. Patients who underwent systemic plus local therapy had significantly better PFS than those who underwent local or systemic therapy alone (*p* < 0.0001, *p* = 0.008, respectively). Detailed information on PFS for each treatment—preoperative therapy plus surgery, chemoradiotherapy, surgery alone, radiotherapy, and chemotherapy—is presented in the Appendix A. Postoperative recurrence was observed in 33 cases (56.9%), including 5 cases in the locoregional area, 21 cases in distant sites, and 7 cases in both.

### 3.7. Relationship Between Number of Metastatic Organs and Clinical Response

Clinical response correlated with prognosis. Overall survival according to clinical response in the entire cohort is shown in Appendix A. The relationship between the number of metastatic organs and clinical response is shown in Table 3. Among patients with one metastatic organ, complete and partial responses were observed in 35 (20.2%) and 91 (52.6%) patients, respectively. For patients with two metastatic organs, complete and partial responses were observed in 0 and 3 patients (23.1%), respectively. For patients with three organ metastases, no treatment responses were observed.

The objective response rates for patients with one, two, or three metastatic organs were 72.8, 23.1, and 0%, respectively. Clinical response was significantly poorer in patients with two or three metastatic organs compared to those with one metastatic organ (*p* = 0.0001).

## 4. Discussion

This study evaluated treatment outcomes in patients with ESCC and synchronous oligometastases. Despite being classified as stage IVB, some patients achieved long-term survival with combined systemic and local therapies. These findings suggest that oligometastatic ESCC may represent a distinct subgroup with potential for improved outcomes through multidisciplinary treatment approaches.

The definition of oligometastases has been widely discussed. According to the consensus documents of the European Society for Radiotherapy and Oncology and the American Society for Radiation Oncology, oligometastatic disease is defined as one to five metastatic lesions that can be safely treated, regardless of primary tumor control [22]. However, these reports were largely based on evidence from colorectal and lung cancers, and the definition of esophageal cancer remains unclear. Subsequently, the OligoMetastatic Esophagogastric Cancer (OMEC) project in Europe defined oligometastasis in esophageal and gastroesophageal cancers as ≤3 metastases in a single organ or metastasis in a single non-regional lymph node station [23]. Nonetheless, questions regarding histological differences, regional disparities, and interinstitutional variations in treatment strategies remain. Establishing a universal definition of oligometastasis is challenging, and it is crucial to gather further evidence.

In this study, we focused on synchronous oligometastatic ESCC. The number of organ metastases was a prognostic factor for OS, and patients with metastases in two or more organs showed markedly poorer treatment response, PFS, and OS. The presence of metastases in two or more organs likely does not represent an intermediate state characteristic of oligometastasis between local disease and widespread dissemination. Similarly to the OMEC project, metastasis confined to a single organ may be a key feature of synchronous oligometastasis in ESCC. However, further multicenter data are required to clearly define synchronous oligometastatic ESCC.

Reports on the treatment outcomes of synchronous oligometastatic esophageal cancer are scarce compared with those of metachronous cases, and studies on ESCC are extremely limited. However, two studies have demonstrated the efficacy of chemoradiotherapy. Shi et al. defined oligometastasis as ≤5 lesions and compared definitive concurrent chemoradiotherapy with chemotherapy alone in oligometastatic cases. The chemoradiotherapy group showed significantly better outcomes than the chemotherapy-only group in objective response rate, PFS, and OS (objective response rate of 57.9% vs. 42.1%; median PFS of 9.7 vs. 7.6 months; median OS of 18.5 vs. 15.2 months) [24]. Chen et al., who included a subset of adenocarcinoma cases, defined oligometastasis as ≤3 lesions and compared chemoradiotherapy with chemotherapy alone in patients with oligometastasis. The median PFS and OS for the entire cohort were 7.8 and 15.9 months, respectively. The chemoradiotherapy group showed significantly improved PFS compared with chemotherapy alone (median PFS of 8.7 vs. 7.3 months), while OS did not differ significantly between groups. While longer-term OS did not differ significantly between groups, longer-term OS was observed in the chemoradiotherapy group (median OS 16.8 vs. 14.8 months) [25]. In our study, the chemoradiotherapy group achieved similar outcomes (median PFS of 6.8 months, OS of 20.2 months; Figure 1 and Appendix A).

Regarding surgical treatment, no reports have specifically focused on synchronous oligometastatic ESCC. Sugimura et al. [26] and Tsuji et al. [27] summarized the outcomes of conversion surgery for esophageal cancer with distant metastases. Although their cohorts were not strictly limited to oligometastatic cases, most patients had metastases confined to a single organ, suggesting few distant lesions. They reported 3-year and 5-year OS rates of 32.4–47.5% and 24.4–31.7%, respectively [26,27]. Similarly, although no study has specifically focused on oligometastasis, Igaue et al. [28] reported outcomes in patients with distant lymph node metastases treated with preoperative chemotherapy followed by surgery, with a 3-year OS rate of 76.7%. In our study, the preoperative therapy plus surgery group achieved a 3-year OS rate of 63.8% and a median OS of 54.3 months, demonstrating favorable outcomes consistent with previous reports (Figure 1).

As outlined above, in oligometastatic cases, combining systemic therapies, such as chemotherapy or immunotherapy, with local therapies, including radiotherapy and surgery, appears to improve outcomes. Norén et al. [29], although focusing on adenocarcinoma, reported that multimodal treatment combining surgical resection and local therapy for metastatic lesions (surgery, chemoradiotherapy, or liver ablation) in patients with synchronous oligometastatic esophageal and gastroesophageal junction cancer resulted in a median OS of 20.9 months. Similarly, Sander et al. [30] reported outcomes in patients with adenocarcinoma with synchronous liver oligometastases treated with surgery plus therapy for liver metastases (resection or ablation), with a median OS of 21 months. In our study, systemic and local therapy (initial therapy plus surgery or chemoradiotherapy) resulted in a median OS of 35.9 months, suggesting that multimodal treatment is an important strategy for synchronous oligometastatic ESCC. However, reports on treatment outcomes for synchronous oligometastatic ESCC remain limited, and there is a need for further data. Advancements in this field are key to improving the prognosis of patients with stage IVB esophageal cancer.

This study has some limitations. First, it was conducted at a single facility and used a retrospective design. Second, the long case registration period, which includes older cases, may introduce bias due to advances in diagnostic methods and treatment content. However, we believe that no difference in prognosis was observed when comparing treatment eras, indicating that this bias is not significant. Third, the emergence of immunotherapy has led to the availability of multiple chemotherapy regimens, resulting in variations in systemic treatment. Fourth, in cases treated with non-surgical methods, pathological evaluation of distant metastatic lesions is difficult. Fifth, treatment outcomes could not be uniformly compared because of potential biases related to background factors, such as systemic condition, comorbidities, and the type and status of distant metastases. Therefore, caution is required when interpreting treatment outcomes. Nevertheless, a comprehensive prognostic analysis was performed, taking these factors into account to minimize bias in outcome interpretation. While determining the optimal treatment strategy requires large multicenter studies, this study provides important insights for improving the prognosis of ESCC with synchronous oligometastases.

## 5. Conclusions

In conclusion, a subset of patients with ESCC and synchronous oligometastases achieved long-term survival after receiving multimodal treatment combining systemic and local therapies. Although patients with multiorgan metastases showed poor outcomes, these findings suggest that selected patients with limited metastases may benefit from an aggressive multidisciplinary approach.

## Figures and Tables

**Figure 1 cancers-17-03407-f001:**
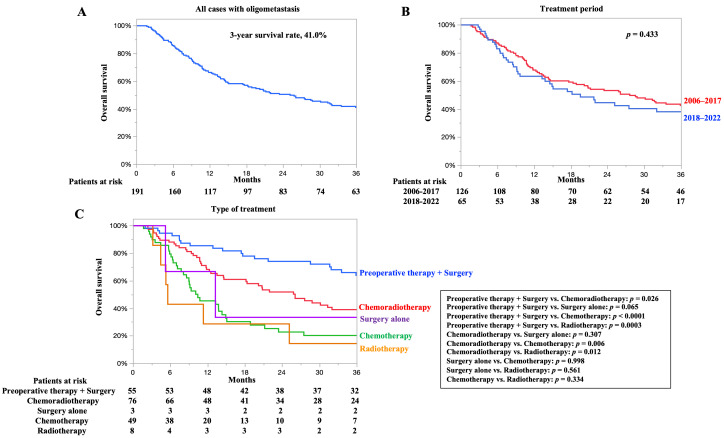
(**A**) Overall survival (OS) for all cases. (**B**) OS according to the treatment period. (**C**) OS according to the type of treatment. Chemotherapy also includes immunotherapy.

**Figure 2 cancers-17-03407-f002:**
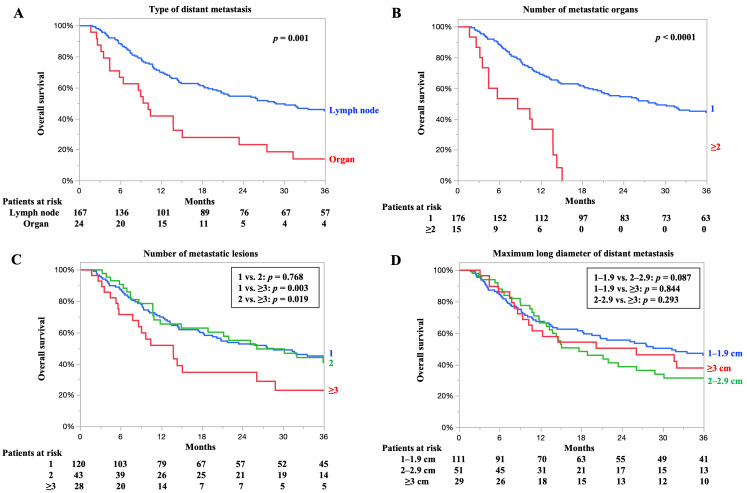
Overall survival (OS) according to factors involved in oligometastasis. (**A**) OS according to type of distant metastasis. Cases exhibiting both organ metastasis and lymph node metastasis were assigned to the organ metastasis group. (**B**) OS according to number of metastatic organs. Distant lymph node metastasis is counted as one organ metastasis, even if there are multiple stations. (**C**) OS according to the number of metastatic lesions. (**D**) OS according to the maximum long diameter of distant metastasis.

**Figure 3 cancers-17-03407-f003:**
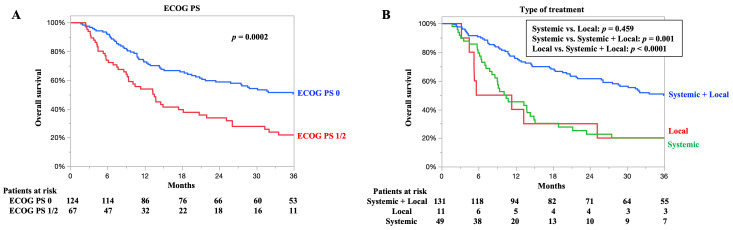
Overall survival (OS) according to independent pretreatment prognostic factors. (**A**) OS according to Eastern Cooperative Oncology Group performance status. (**B**) OS according to type of treatment. The treatment content was broadly divided into the following three categories: systemic therapy plus local therapy (initial therapy and surgery, chemoradiotherapy), local therapy (radiotherapy, surgery alone), and systemic therapy (chemotherapy, immunotherapy).

**Figure 4 cancers-17-03407-f004:**
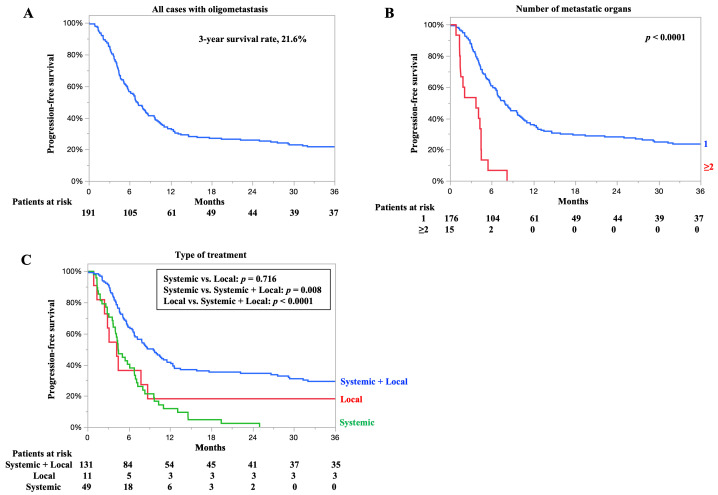
Progression-free survival (PFS) for all cases and according to the number of metastatic organs and type of treatment. (**A**) PFS for all cases. (**B**) PFS according to the number of metastatic organs. (**C**) PFS according to type of treatment. The treatment content was broadly divided into the following three categories: systemic therapy plus local therapy (initial therapy and surgery, chemoradiotherapy), systemic therapy (chemotherapy, immunotherapy), and local therapy (radiotherapy, surgery alone).

**Table 1 cancers-17-03407-t001:** Clinicopathologic characteristics of patients with synchronous oligometastasis.

Parameters	N = 191
Age (mean ± SD, y)	68.2 ± 10.1
Sex	
Male	169 (88.5%)
Female	22 (11.5%)
ECOG PS	
0	124 (64.9%)
1	62 (32.5%)
2	5 (2.6%)
Tumor markers	
SCC (mean ± SD, ng/mL)	2.2 ± 1.9
CEA (mean ± SD, ng/mL)	3.9 ± 3.9
Primary tumor location	
Upper	64 (33.5%)
Middle	82 (42.9%)
Lower	42 (22.0%)
Abdominal	3 (1.6%)
Histology (biopsy specimens)	
Well-differentiated	13 (6.8%)
Moderately differentiated	30 (15.7%)
Poorly differentiated	38 (19.9%)
Squamous cell carcinoma (not assessable)	110 (57.6%)
Clinical T ^a^	
cT1	7 (3.7%)
cT2	15 (7.9%)
cT3	116 (60.7%)
cT4	53 (27.7%)
Clinical N ^a^	
cN0	30 (15.7%)
cN1	81 (42.4%)
cN2	74 (38.8%)
cN3	6 (3.1%)
Details of distant metastasis ^b^	
Distant lymph node	175 (92.1%)
Liver	14 (7.3%)
Bone	8 (4.2%)
Lung	6 (3.1%)
Adrenal gland	2 (1.1%)
Skeletal muscle	1 (0.5%)
Type of distant metastasis ^c^	
Organ	24 (12.6%)
Lymph node	167 (87.4%)
Number of metastatic organs ^d^	
1	176 (92.1%)
2	13 (6.8%)
3	2 (1.1%)
Number of metastatic lesions	
1	120 (62.8%)
2	43 (22.5%)
3	19 (10.0%)
4	8 (4.2%)
5	1 (0.5%)
Maximum long diameter of distant metastasis	
(mean ± SD, cm)	2.0 ± 0.9
1–1.9 cm	111 (58.1%)
2–2.9 cm	51 (26.7%)
3–3.9 cm	19 (9.9%)
4–4.9 cm	7 (3.7%)
5–5.9 cm	3 (1.6%)
Type of treatment	
Systemic + local therapy	
Preoperative chemotherapy and surgery	26 (13.6%)
Preoperative chemoradiotherapy and surgery	29 (15.2%)
Chemoradiotherapy	76 (39.8%)
Local therapy	
Surgery alone	3 (1.6%)
Radiotherapy	8 (4.2%)
Systemic therapy ^e^	
Chemotherapy + immunotherapy	7 (3.6%)
Chemotherapy	39 (20.4%)
Immunotherapy	3 (1.6%)
Clinical response ^f,g^	
Complete response	35 (18.6%)
Partial response	94 (50.0%)
Stable disease	18 (9.6%)
Progressive disease	37 (19.6%)
No assessment	4 (2.2%)

Values are shown as n (%) or mean ± SD. SD, standard deviation. ^a^ Clinical staging according to TNM classification, 8th edition. ^b^ Cases with multiple types of distant metastases are also included. ^c^ Cases exhibiting both organ metastasis and lymph node metastasis were assigned to the organ metastasis group. The lymph node group includes cases with distant lymph node metastases only. ^d^ Distant lymph node metastasis is counted as one organ, even if there are multiple stations. ^e^ Chemotherapy also includes immunotherapy. ^f^ According to the Response Evaluation Criteria in Solid Tumors (RECIST). ^g^ Clinical response in cases, excluding the three cases treated with surgery alone. Abbreviations: ECOG PS, Eastern Cooperative Oncology Group performance status; SCC, squamous cell carcinoma-related antigen; CEA, carcinoembryonic antigen.

**Table 2 cancers-17-03407-t002:** Results of univariate and multivariate analyses of prognostic factors for overall survival.

	Univariate			Multivariate		
Variables	HR	95% CI	*p*-Value	HR	95% CI	*p*-Value
Age, y (continuous)	1.02	0.98–1.04	0.978	–	–	–
Sex						
Female (reference)	1			–	–	–
Male	1.19	0.68–2.07	0.535	–	–	–
ECOG PS						
0 (reference)	1			1		
1/2	1.99	1.38–2.86	0.0002	1.52	1.02–2.27	0.036
SCC ^a^						
<1.5 ng/mL	1			1		
≥1.5 ng/mL	1.50	1.05–2.14	0.024	1.39	0.95–2.02	0.084
CEA ^a^						
<5 ng/mL	1			–	–	–
≥5 ng/mL	1.50	0.96–2.33	0.071	–	–	–
Primary tumor location						
Lower, abdominal (reference)	1			–	–	–
Upper, middle	1.01	0.66–1.54	0.956	–	–	–
Histology (biopsy specimens)						
Others (reference)	1			–	–	–
Poorly differentiated	1.04	0.56–1.59	0.856	–	–	–
Clinical T ^b^						
T1/2 (reference)	1			–	–	–
T3/4	1.42	0.82–2.44	0.199	–	–	–
Clinical N ^b^						
N0/1 (reference)	1			–	–	–
N2/3	1.21	0.84–1.73	0.305	–	–	–
Type of distant metastasis ^c^						
Lymph node (reference)	1					
Organ	2.27	1.39–3.68	0.001	1.14	0.57–2.24	0.702
Number of metastatic organs ^d^						
1 (reference)	1			1		
≥2	4.23	2.35–7.60	<0.0001	2.72	1.09–6.78	0.031
Number of metastatic lesions						
1 (reference)	1			1		
2	1.06	0.69–1.64	0.768	1.05	0.66–1.68	0.815
≥3	2.09	1.29–3.38	0.003	1.19	0.61–2.33	0.599
Maximum long diameter of distant metastasis						
1–1.9 cm (reference)	1			–	–	–
2–2.9 cm	1.41	0.95–2.11	0.087	–	–	–
≥3 cm	1.05	0.63–1.74	0.844	–	–	–
Type of treatment ^e^						
Systemic therapy (reference)	1			1		
Local therapy	1.30	0.64–2.63	0.459	1.26	0.59–2.73	0.541
Systemic + local therapy	0.43	0.29–0.65	<0.0001	0.56	0.34–0.93	0.026
Treatment period						
2006–2017	1			–	–	–
2018–2022	1.16	0.79–1.71	0.433	–	–	–

^a^ The cutoffs for pretherapeutic CEA and SCC were set at the normal values of 5.0 and 1.5 ng/mL, respectively. ^b^ Clinical staging according to TNM classification, 8th edition. ^c^ Cases exhibiting both organ metastasis and lymph node metastasis were assigned to the organ metastasis group. ^d^ Distant lymph node metastasis is counted as one organ, even if there are multiple stations. ^e^ Systemic therapy plus local therapy (initial therapy and surgery, chemoradiotherapy), local therapy (radiotherapy, surgery alone), systemic therapy (chemotherapy, immunotherapy). HR, hazard ratio; CI, confidence interval; ECOG PS, Eastern Cooperative Oncology Group performance status; SCC, squamous cell carcinoma-related antigen; CEA, carcinoembryonic antigen.

**Table 3 cancers-17-03407-t003:** Clinical response according to the number of metastatic organs.

Clinical Response ^a,b^	N = 191	Number of Metastatic Organs ^c^			*p*-Value
		1 (n = 173)	2 (n = 13)	3 (n = 2)	
Complete response	35 (18.6%)	35 (20.2%)	0	0	0.0001
Partial response	94 (50.0%)	91 (52.6%)	3 (23.1%)	0	
Stable disease	18 (9.6%)	17 (9.9%)	1 (7.7%)	0	
Progressive disease	37 (19.6%)	26 (15.0%)	9 (69.2%)	2 (100%)	
No assessment	4 (2.2%)	4 (2.3%)	0	0	

^a^ According to the Response Evaluation Criteria in Solid Tumors (RECIST). ^b^ Clinical response in cases excluding the three cases treated with surgery alone. ^c^ Distant lymph node metastasis is counted as one organ, even if there are multiple stations.

## Data Availability

Data presented in this study are available from the corresponding author upon request. The data are not publicly available for further analysis.

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
