# Peer review of "Treatment Outcomes and Significance of Multimodal Treatment in Esophageal Squamous Cell Carcinoma with Synchronous Oligometastasis"

_cancers, 2025, doi:10.3390/cancers17213407_

Round 1

Reviewer 1 Report

Comments and Suggestions for Authors

The long accrual period of 16 years introduces an unavoidable bias as diagnostics and treatment modalities have evolved over such long time. E.g. immunotherapy was not yet used until recently, some markers, such as the CPS, not yet known. This point should be addressed by the authors.

Similarly, the authors mentioned that the 8th edition of the TNM classification (2016) was used. Does this mean that patients treated before were re-classified according to the 8th edition?

What was the incidence of oligometastatic patients compared to all patients with metastasis during the inclusion period? And had the introduction of PET-CT an influence on the detection rate of oligometastatic patients?

Table 1: what is the difference between “details of distant metastasis” and “type of distant metastasis?

Was there a difference regarding the long-term outcome of patients with complete response versus partial response versus progressive disease?

The assessment of treatment response included only the distant metastasis or also the primary tumor?

As distant metastasis were not systematically resected, there is no pathological confirmation of the treatment response. Please comment.

There were six different types of systemic treatment, and many of them were only used in less than 7 patients. Obviously, this implicates some bias and impairs the interpretation of the treatment effect.

Only 58 patients out of 191 patients were considered as surgical candidates. How were they selected?

Discussion section

The discussion should start with 1-2 sentence short summary of the main findings. The actual version remains vague. The conclusion that this study will be important to improve the treatment outcome of oligometastatic ESSC is not based on study findings.

Bias must be more clearly addressed, and not only in a “proforma” way.

It is not very probable that for such a rare clinical condition, prospective studies will be performed (which is in fact a real shortcoming). Hence, this proposition should be replaced.

The conclusions must be rephrased. The conclusions are not a summary of the study.

Author Response

Reviewer 1:

Comments and Suggestions for Authors

The long accrual period of 16 years introduces an unavoidable bias as diagnostics and treatment modalities have evolved over such long time. E.g. immunotherapy was not yet used until recently, some markers, such as the CPS, not yet known. This point should be addressed by the authors.

Response:

As you pointed out, the long case registration period and inclusion of older cases may introduce bias due to advances in diagnostic methods and treatment. We have added the following text to the “Discussion” section:

The long case registration period, which includes older cases, may introduce bias due to advances in diagnostic methods and treatment content. However, we believe that no difference in prognosis was observed when comparing treatment eras, indicating that this bias is not significant.

Additionally, OS curves by treatment period (2006-2017 vs. 2018-2022) have been added to Figure 1. As depicted in Figure 1, no difference in prognosis was observed across the two treatment periods. Furthermore, we re-performed the prognostic analysis in Table 2 after including treatment period as a factor, and no correlation with prognosis was observed.

Similarly, the authors mentioned that the 8th edition of the TNM classification (2016) was used. Does this mean that patients treated before were re-classified according to the 8th edition?

Response:

Cases older than 2016 have also been reclassified and evaluated using the TNM Classification, 8th edition. The following sentence has been added to the “Patients” section:

Cases older than 2016 have also been reclassified and evaluated using the 8th edition.

What was the incidence of oligometastatic patients compared to all patients with metastasis during the inclusion period? And had the introduction of PET-CT an influence on the detection rate of oligometastatic patients?

Response:

We investigated the incidence of oligometastases and added the following text to the “Details of Oligometastases and Treatments” section.

During the registration period, there were 258 stage IVB cases, and the incidence of oligometastases was 74.0%. The incidence of oligometastases before and after the introduction of ¹⁸F-fluorodeoxyglucose positron emission tomography in 2008 was 61.5% and 75.4%, respectively, showing no significant difference.

Table 1: what is the difference between “details of distant metastasis” and “type of distant metastasis?

Response:

In Table 1, “Details of distant metastasis” refer to the number of cases corresponding to each type of metastasis (some cases may be duplicated; for example, a case with both distant lymph node and lung metastasis is counted twice). “Type of distant metastasis” indicates whether a case has organ or only distant lymph node metastasis (there are no duplicate cases. For example, a case with both distant lymph node metastasis and lung metastasis is counted in the organ group). To clarify, the following sentence has been added to annotation c in Table 1.

The lymph node group includes cases with distant lymph node metastases only.

Was there a difference regarding the long-term outcome of patients with complete response versus partial response versus progressive disease?

Response:

Differences in prognosis were observed based on the treatment response. We have created OS curves for each treatment response category as a supplementary figure. The following text has been added to the section “Relationship between Number of Metastatic Organs and Clinical Response,” and the figure has been added to the supplementary file.

Clinical response correlates with prognosis. Overall survival according to the clinical response in the entire cohort is shown in Supplementary Data (Figure S2).

The assessment of treatment response included only the distant metastasis or also the primary tumor?

Response:

The assessment was based on RECIST. We have accordingly added the following text in the “Patients” section:

Based on the Response Evaluation Criteria in Solid Tumors, the primary tumor was an unmeasurable lesion, and treatment response was evaluated based on measurable lesions among lymph node and distant metastases.

As distant metastasis were not systematically resected, there is no pathological confirmation of the treatment response. Please comment.

Response:

In cases treated with non-surgical methods, diagnosis was based solely on imaging, leaving it unclear whether cancer truly existed in the distant metastatic sites. However, as noted in the “Oligometastasis” section, imaging assessments were evaluated by two independent radiologists, and biopsy was performed whenever possible for cases that were difficult to assess by imaging alone. This remains an unavoidable limitation, and the following sentence has been added to the “Discussion” section:

In cases treated with non-surgical methods, pathological evaluation of distant metastatic lesions is difficult.

There were six different types of systemic treatment, and many of them were only used in less than 7 patients. Obviously, this implicates some bias and impairs the interpretation of the treatment effect.

Response:

Although bias may exist, the emergence of immunotherapy in Japan has led to the availability of multiple chemotherapy regimens for Stage IV esophageal cancer, making variations in treatment regimens unavoidable. This limitation has been added to the “Discussion” section as follows:

The emergence of immunotherapy has led to the availability of multiple chemotherapy regimens, resulting in variations in systemic treatment.

Only 58 patients out of 191 patients were considered as surgical candidates. How were they selected?

Response:

As stated in the “Treatment Selection” section, the primary requirement was that the patient was fit for surgery. Another criterion, as noted in the “Surgical Treatment” section, was that cases where the organ metastases disappeared on imaging after preoperative treatment were considered surgically eligible. For distant lymph node metastases, cases involving metastases in areas relatively close to the esophagus (such as lymph nodes in the thoracic pretracheal region) were also considered surgically eligible. This information has been added to the “Treatment Selection” section. The text now reads as follows:

For organ metastases, cases where organ metastases disappeared on imaging after preoperative therapy were considered suitable for surgery. For distant lymph node metastases, cases involving metastases in areas relatively close to the esophagus (such as lymph nodes in the thoracic pretracheal region) were considered suitable for surgery.

Discussion section

The discussion should start with 1-2 sentence short summary of the main findings. The actual version remains vague. The conclusion that this study will be important to improve the treatment outcome of oligometastatic ESSC is not based on study findings.

Response:

We have revised the text as follows:

This study evaluated treatment outcomes in patients with ESCC and synchronous oligometastases. Despite being classified as stage IVB, some patients achieved long-term survival with combined systemic and local therapies. These findings suggest that oligometastatic ESCC may represent a distinct subgroup with potential for improved outcomes through multidisciplinary treatment approaches.

Bias must be more clearly addressed, and not only in a “proforma” way.

Response: 

We have revised the sentence as follows:

Treatment outcomes could not be uniformly compared because of potential biases related to background factors, such as systemic condition, comorbidities, and the type and status of distant metastases. Therefore, caution is required when interpreting treatment outcomes. Nevertheless, a comprehensive prognostic analysis was performed, taking these factors into account to minimize bias in outcome interpretation.

It is not very probable that for such a rare clinical condition, prospective studies will be performed (which is in fact a real shortcoming). Hence, this proposition should be replaced.

Response:

We have revised the text as follows:

While determining the optimal treatment strategy requires large multicenter studies, this study provides important insights for improving the prognosis of ESCC with synchronous oligometastases.

The conclusions must be rephrased. The conclusions are not a summary of the study.

Response:

We have rewritten the conclusions as follows:

In conclusion, a subset of patients with ESCC and synchronous oligometastases achieved long-term survival through multimodal treatment combining systemic and local therapies. Although patients with multiorgan metastases showed poor outcomes, these findings suggest that selected patients with limited metastases may benefit from an aggressive multidisciplinary approach.

Reviewer 2 Report

Comments and Suggestions for Authors

The authors analyze 191 patients with ESCC and synchronous oligometastasis to identify prognostic factors and the treatment strategy associated with the most favorable outcomes. Although a multimodal (systemic + local) approach is clinically hypothesized to yield the best results, most single centers struggle to accrue enough such cases to demonstrate this convincingly. In that regard, this study is valuable and helps address an evidence gap. That said, several minor issues should be corrected or clarified.

Minor comments

  1. In Table 1, the mean of the maximum long diameter (20.3 cm) appears to be an error.
  2. The Table 1 footnote (“e. Chemotherapy also includes immunotherapy.”) and the Figure 1 caption (“The chemotherapy group is included in the immunotherapy group.”) could cause confusion and should be checked and revised.
  3. In 3.4 Univariate and Multivariate Analyses of OS, the directions of the inequality signs appear to be incorrect and need verification.
  4. Most patients would likely have been evaluated using RECIST 1.1 (2009), but the cited reference is the earlier 2000 version.

Author Response

Reviewer 2

Comments and Suggestions for Authors

The authors analyze 191 patients with ESCC and synchronous oligometastasis to identify prognostic factors and the treatment strategy associated with the most favorable outcomes. Although a multimodal (systemic + local) approach is clinically hypothesized to yield the best results, most single centers struggle to accrue enough such cases to demonstrate this convincingly. In that regard, this study is valuable and helps address an evidence gap. That said, several minor issues should be corrected or clarified.

Minor comments

In Table 1, the mean of the maximum long diameter (20.3 cm) appears to be an error.

Response:

We have corrected for the mistake regarding the mean having a value of 2.3 cm (20.3 mm).

The Table 1 footnote (“e. Chemotherapy also includes immunotherapy.”) and the Figure 1 caption (“The chemotherapy group is included in the immunotherapy group.”) could cause confusion and should be checked and revised.

Response:

To avoid confusion, the text in Figure 1 has been revised to match the information in Table 1.

In 3.4 Univariate and Multivariate Analyses of OS, the directions of the inequality signs appear to be incorrect and need verification.

Response:

The inequality sign has been corrected.

Most patients would likely have been evaluated using RECIST 1.1 (2009), but the cited reference is the earlier 2000 version.

Response:

Our evaluations were performed using RECIST 1.1 (2009). We have revised the cited reference.

Reviewer 3 Report

Comments and Suggestions for Authors

This retrospective study investigates an important and clinically relevant question: whether patients with stage IVB esophageal squamous cell carcinoma (ESCC) and synchronous oligometastases represent a distinct subgroup that may benefit from aggressive multimodal therapy. The authors define oligometastasis as ≤5 metastatic lesions and analyze survival outcomes in a cohort of 191 patients treated over a 16-year period. The central finding—that combined systemic and local therapy is associated with significantly improved survival, particularly in patients with single-organ metastasis—adds valuable evidence to the evolving concept of oligometastatic disease in aggressive solid tumors. But I have several following concerns: 

1. The authors adopt the commonly used definition of ≤5 metastatic lesions. However, recent literature increasingly emphasizes metastatic burden, distribution (single vs. multiple organs), and biological behavior as equally or more important than lesion count alone. The striking difference in survival between single-organ (3-year OS: 44.3%) and multiple-organ metastasis (0%) strongly suggests that "number of involved organs" may be a more robust discriminator than lesion number. The authors should consider refining the definition of "true" oligometastasis in their cohort based on organ involvement and discuss whether lesion count alone is sufficient for patient stratification.

2. The treatment groups (systemic, local, and combined) are heterogeneous in terms of modalities (e.g., surgery vs. radiotherapy), timing, and systemic regimens (especially with the inclusion of immunotherapy in more recent years). Given that immunotherapy has significantly altered the landscape of ESCC treatment post-2018, its impact on survival should be addressed. Was there a temporal trend in outcomes (e.g., patients treated after 2018 vs. before)? A subgroup analysis by treatment era or immunotherapy use would strengthen the validity of the conclusions.

3. As this is a retrospective study, patients selected for local therapy (especially surgery) likely had better performance status, fewer comorbidities, and more favorable disease biology. While multivariate analysis adjusts for some factors, residual confounding cannot be ruled out. The authors should explicitly acknowledge this limitation and discuss how patient selection may have influenced the observed survival benefit in the combined therapy group.

4. While overall survival and progression-free survival are primary endpoints, information on local control rates, patterns of recurrence, and treatment-related toxicity (especially for combined modality therapy) would provide a more complete picture of the risk-benefit ratio. For example, did patients receiving local therapy experience fewer local or distant failures? Were there differences in treatment-related mortality or morbidity?

5. Please clarify whether "preoperative therapy" refers to chemotherapy or chemoradiotherapy, and whether the radiation doses and systemic regimens were consistent across the cohort.

6. Consider adding a Kaplan-Meier curve (if not already in the full manuscript) to visually represent the survival differences between treatment groups.

7. The term "dismal outcomes" for multiple-organ metastases is accurate but could be softened to "very poor prognosis" for a more neutral tone.

8. Define "performance status" (e.g., ECOG or Karnofsky) in the methods section.

9. Please unify the format of references in the article, including the author's name, the case of words in the title of the article, the writing of the name of the journal, and the page number.

Comments on the Quality of English Language

The English could be improved to more clearly express the research.

Author Response

Reviewer 3

Comments and Suggestions for Authors

This retrospective study investigates an important and clinically relevant question: whether patients with stage IVB esophageal squamous cell carcinoma (ESCC) and synchronous oligometastases represent a distinct subgroup that may benefit from aggressive multimodal therapy. The authors define oligometastasis as ≤5 metastatic lesions and analyze survival outcomes in a cohort of 191 patients treated over a 16-year period. The central finding—that combined systemic and local therapy is associated with significantly improved survival, particularly in patients with single-organ metastasis—adds valuable evidence to the evolving concept of oligometastatic disease in aggressive solid tumors. But I have several following concerns:

  1. The authors adopt the commonly used definition of ≤5 metastatic lesions. However, recent literature increasingly emphasizes metastatic burden, distribution (single vs. multiple organs), and biological behavior as equally or more important than lesion count alone. The striking difference in survival between single-organ (3-year OS: 44.3%) and multiple-organ metastasis (0%) strongly suggests that "number of involved organs" may be a more robust discriminator than lesion number. The authors should consider refining the definition of "true" oligometastasis in their cohort based on organ involvement and discuss whether lesion count alone is sufficient for patient stratification.

Response:

Thank you for your valuable feedback. As noted in the “Discussion” section, while the definition of oligometastasis is becoming more established based on results from projects like OMEC, the data primarily involve adenocarcinomas. Data on oligometastasis in ESCC remain limited, and we believe that a definition for this patient population needs to be established as well. In this study, we obtained data suggesting that a single organ metastasis may be the defining point for oligometastasis in ESCC. However, the data from this study alone are insufficient to clearly refine the definition, and further multicenter data are required. We have added the following sentence to the “Discussion” section:

However, further multicenter data are required to clearly define synchronous oligometastatic ESCC.

  1. The treatment groups (systemic, local, and combined) are heterogeneous in terms of modalities (e.g., surgery vs. radiotherapy), timing, and systemic regimens (especially with the inclusion of immunotherapy in more recent years). Given that immunotherapy has significantly altered the landscape of ESCC treatment post-2018, its impact on survival should be addressed. Was there a temporal trend in outcomes (e.g., patients treated after 2018 vs. before)? A subgroup analysis by treatment era or immunotherapy use would strengthen the validity of the conclusions.

Response:

We have added OS curves by treatment period (2006-2017 vs. 2018-2022) to Figure 1. No difference in prognosis was observed based on the treatment period. Furthermore, we reran the prognostic analysis in Table 2 after including the treatment period as a factor. No correlation with prognosis was observed.

  1. As this is a retrospective study, patients selected for local therapy (especially surgery) likely had better performance status, fewer comorbidities, and more favorable disease biology. While multivariate analysis adjusts for some factors, residual confounding cannot be ruled out. The authors should explicitly acknowledge this limitation and discuss how patient selection may have influenced the observed survival benefit in the combined therapy group.

Response:

We made every effort to minimize bias due to patient background in the multivariate analysis, but we also recognize that it cannot be completely eliminated. We have added the following sentence as a limitation in the “Discussion” section:

Treatment outcomes could not be uniformly compared because of potential biases related to background factors, such as systemic condition, comorbidities, and the type and status of distant metastases. Therefore, caution is required when interpreting treatment outcomes. Nevertheless, a comprehensive prognostic analysis was performed, taking these factors into account to minimize bias in outcome interpretation.

  1. While overall survival and progression-free survival are primary endpoints, information on local control rates, patterns of recurrence, and treatment-related toxicity (especially for combined modality therapy) would provide a more complete picture of the risk-benefit ratio. For example, did patients receiving local therapy experience fewer local or distant failures? Were there differences in treatment-related mortality or morbidity?

Response:

The following has been added to the “PFS According to All Cases, Number of Metastatic Organs, and Type of Treatment” section regarding postoperative recurrence rates and recurrence patterns:

Postoperative recurrence was observed in 33 cases (56.9%): 5 cases in the locoregional area, 21 cases in distant sites, and 7 cases in both.

Regarding treatment-related mortality, the following has been added to the “Details of Oligometastases and Treatments” section:

Treatment-related deaths included one case following chemoradiotherapy and another following chemotherapy.

  1. Please clarify whether "preoperative therapy" refers to chemotherapy or chemoradiotherapy, and whether the radiation doses and systemic regimens were consistent across the cohort.

Response:

Preoperative therapy refers to both chemotherapy and chemoradiotherapy. The text in the “Treatment Selection” section has been revised as follows:

Surgery is generally performed after preoperative therapy, which included either chemotherapy or chemoradiotherapy.

Regarding systemic therapy and radiotherapy, the following text has been added to the “Chemotherapy and Immunotherapy” and “Definitive Chemoradiotherapy and Radiotherapy” sections:

Chemotherapy regimens were generally delivered at standard doses, with adjustments made on an individual basis depending on treatment-related toxicities.

The prescribed radiation doses were uniform throughout the cohort.

  1. Consider adding a Kaplan-Meier curve (if not already in the full manuscript) to visually represent the survival differences between treatment groups.

Response:

Survival curves for the treatment groups are presented in Figure 1, Figure 3, Figure 4, and Figure S1. Please refer to these figures.

  1. The term "dismal outcomes" for multiple-organ metastases is accurate but could be softened to "very poor prognosis" for a more neutral tone.

Response:

We have made the correction as suggested.

  1. Define "performance status" (e.g., ECOG or Karnofsky) in the methods section.

Response:

The following text has been added to the “Patients” section:

Performance status was assessed using the Eastern Cooperative Oncology Group performance status (ECOG PS) scale.

  1. Please unify the format of references in the article, including the author's name, the case of words in the title of the article, the writing of the name of the journal, and the page number.

Response:

We have formatted the references in accordance with the journal guidelines.

Reviewer 4 Report

Comments and Suggestions for Authors

This retrospective study examines outcomes of multimodal versus single-modality treatments in 191 patients with esophageal squamous cell carcinoma (ESCC) and synchronous oligometastases treated at Hiroshima University between 2006 and 2022. The study addresses an important and understudied clinical question, providing valuable long-term survival data and identifying prognostic factors. However, there are some issues and questions that need to be addressed in the study.

1- The article defines oligometastasis as ≤5 distant lesions, but this choice is not sufficiently justified. Given recent OMEC and ESTRO–ASTRO consensus documents, authors should provide a clear rationale for their definition and discuss how their criteria align with or differ from current standards. It would be helpful to present a subgroup analysis using alternative definitions (e.g., ≤3 lesions or single-organ metastasis only) to assess the robustness of the findings.

2- The article should present baseline characteristics stratified by treatment group, including ECOG, number and location of metastases, and clinical stage, to clarify whether differences in survival are attributable to selection rather than treatment effect.

3- The "Systemic + Local Therapy" group includes both preoperative treatment followed by surgery and definitive chemoradiotherapy, which differ in terms of biologics and prognosis. Separate analyses for these subgroups would be valuable. Similarly, systemic therapy includes chemotherapy, immunotherapy, or both. More detailed reporting (e.g., proportion receiving immune checkpoint inhibitors) would help contextualize the results.

4- Stratifying survival outcomes by treatment period (e.g., pre-2018 and post-2018) would strengthen the article.

5- Variables such as "SCC ≥1.5 ng/mL" and "CEA ≥5 ng/mL" were dichotomized; explain the cutoff values ​​and their clinical basis.

Author Response

Review 4.

This retrospective study examines outcomes of multimodal versus single-modality treatments in 191 patients with esophageal squamous cell carcinoma (ESCC) and synchronous oligometastases treated at Hiroshima University between 2006 and 2022. The study addresses an important and understudied clinical question, providing valuable long-term survival data and identifying prognostic factors. However, there are some issues and questions that need to be addressed in the study.

1- The article defines oligometastasis as ≤5 distant lesions, but this choice is not sufficiently justified. Given recent OMEC and ESTRO–ASTRO consensus documents, authors should provide a clear rationale for their definition and discuss how their criteria align with or differ from current standards. It would be helpful to present a subgroup analysis using alternative definitions (e.g., ≤3 lesions or single-organ metastasis only) to assess the robustness of the findings.

Response:

As noted in the “Introduction” and “Discussion” sections, the definitions in the OMEC and ESTRO–ASTRO consensus documents represent consensus for adenocarcinoma, not squamous cell carcinoma. Therefore, large-scale data analysis, review, and consensus formation are necessary going forward. Consequently, it is necessary to consider slightly broader eligibility criteria, and we believe this paper will be helpful and useful in that regard. The following sentence has been added to the “Discussion” section:

However, further multicenter data are required to clearly define synchronous oligometastatic ESCC.

2- The article should present baseline characteristics stratified by treatment group, including ECOG, number and location of metastases, and clinical stage, to clarify whether differences in survival are attributable to selection rather than treatment effect.

Response:

No statistically significant differences were observed at baseline by treatment. The following sentence has been added to the “Details of Oligometastases and Treatments” section:

No statistically significant differences were observed in baseline characteristics (ECOG PS, Clinical T, Clinical N, number of metastatic organs, number of metastatic lesions) across the three treatment strategies.

3- The "Systemic + Local Therapy" group includes both preoperative treatments followed by surgery and definitive chemoradiotherapy, which differ in terms of biologics and prognosis. Separate analyses for these subgroups would be valuable. Similarly, systemic therapy includes chemotherapy, immunotherapy, or both. More detailed reporting (e.g., proportion receiving immune checkpoint inhibitors) would help contextualize the results.

Response:

For details on the prognosis of preoperative therapy + surgery versus chemoradiotherapy, please refer to Figure 1 and Supplementary Figure S1. As you mentioned, systemic therapy was categorized into three groups: chemotherapy + immunotherapy, chemotherapy alone, and immunotherapy alone. The breakdown is shown in Table 1.

4- Stratifying survival outcomes by treatment period (e.g., pre-2018 and post-2018) would strengthen the article.

Response:

OS curves by treatment period (2006-2017 vs. 2018-2022) have been added to Figure 1.

5- Variables such as "SCC ≥1.5 ng/mL" and "CEA ≥5 ng/mL" were dichotomized; explain the cutoff values and their clinical basis.

Response:

The cutoff values for tumor markers were determined to be within the normal range. The following sentence has been added to Table 2:

The cutoffs for pretherapeutic CEA and SCC were set at the normal values of 5.0 and 1.5 ng/mL, respectively.

Round 2

Reviewer 1 Report

Comments and Suggestions for Authors

The authors answered most raised issues, and overall, the paper improved. No further improvement is probably possible. 

Of importance, the authors now stated that the interpretation of the results must be done with caution considering the inherent bias.

Reviewer 3 Report

Comments and Suggestions for Authors

The authors have addressed all my concerns, I recommend accepting it in current form.

Comments on the Quality of English Language

The English is fine and does not require any improvement.

Reviewer 4 Report

Comments and Suggestions for Authors

I am satisfied that the authors have addressed all of my previous concerns about the article. It is now much improved and I feel that it is now suitable for publication.